# Unique Morphology of *Sarcobatus baileyi* Male Inflorescence and Its Botanical Implications

**DOI:** 10.3390/plants12091917

**Published:** 2023-05-08

**Authors:** Wenzhe Liu, Xiuping Xu, Xin Wang

**Affiliations:** 1Key Laboratory of Resource Biology and Biotechnology in Western China (Ministry of Education), School of Life Sciences, Northwest University, Xi’an 710069, China; lwenzhe@nwu.edu.cn; 2State Key Laboratory of Systematic and Evolutionary Botany, Institute of Botany, Chinese Academy of Sciences, Beijing 100093, China; xpxu@ibcas.ac.cn; 3State Key Laboratory of Palaeobiology and Stratigraphy, Nanjing Institute of Geology and Palaeontology and CAS Center for Excellence in Life and Paleoenvironment, Chinese Academy of Sciences, Nanjing 210008, China

**Keywords:** morphology, North America, ODC (offspring development conditioning), evolution

## Abstract

A typical angiosperm flower is usually bisexual, with entomophilous plants having four whorls of organs: the calyx, corolla, stamens, and gynoecium. The flower is usually colorful, and thus, distinct from the dull-colored reproductive organs of gymnosperms; however, this formula is not applicable to all flowers. For example, the male flower of *Sarcobatus baileyi* is reduced into only a single stamen. Such unusual flowers are largely poorly documented and underappreciated. To fill such a lacuna in our knowledge of the male reproductive organ of *S. baileyi*, we collected and studied materials of the male inflorescence of *S. baileyi* (Sarcobataceae). The outcomes of our Micro-CT (micro computed tomography), SEM (scanning electron microscopy), and paraffin sectioning indicate that a male inflorescence of *S. baileyi* is more comparable with the cone of conifers; its male flowers lack the perianth, are directly attached to a central axis and sheltered by peltate indusium-like shields. To understand the evolutionary logic underlying such a rarely seen male inflorescence, we also studied and compared it with a female cone of *Cupressus sempervirens*. Although the genera *Sarcobatus* and *Cupressus* belong to two distinct major plant groups (angiosperms and gymnosperms), they apply the same propagule-protecting strategy.

## 1. Introduction

Angiosperms are characterized by their flowers, which are frequently a focus of studies in botanical research (including morphological [1,2,3,4,5,6,7,8,9,10,11,12,13,14,15,16,17,18,19,20,21,22] and molecular research [23,24,25,26,27,28]). Most of the flowers in the assumed basal angiosperms (either Magnoliales in former system [29] or ANITA clade in the APG (angiosperm phylogeny group) system [30]) are bisexual. This bisexuality of the reproductive organs in angiosperms constitutes a drastic contrast against the unisexuality of the reproductive organs in gymnosperms (the only sister group of angiosperms in seed plants). In light of this contrast, male flowers in angiosperms seem more likely to bridge the morphological gap between angiosperms and gymnosperms, as female reproductive organs in these two groups are distinct and very hard to homologize. Many male flowers of angiosperms are not purely unisexual; they still have residual gynoecium [31,32]. This makes the pure male flowers in *S. baileyi* especially meaningful in addressing the evolution of male reproductive organs in angiosperms. This prompts us to choose *S. baileyi* as a study material.

*Sarcobatus* (Sarcobataceae) is a genus established by Nees in 1839 based on materials collected by Wied in July, 1833 [33]. It used to be called *Fremontia*, *Sarcacanthus*, *Sarcobatis*, pulpy-leaved thorn, salt cedar, sage, greasewood, and black greasewood by various authors and local peoples [33]. It was placed in Urticeae and Chenopodiaceae by various authors before it was placed in its own monotypic family Sarcobataceae [33]. APG clusters Sarcobataceae, Gieskiaceae, Phytolaccaceae, and Nyctaginaceae together as a clade in Caryophyllales [30]. Different from the other three families in the clade, Sarcobataceae have unisexual flowers and P3cf-form sieve-element plastids, distinguishing them from the other three families in the clade and justifying a separate family, which is also supported by cpDNA sequencing [33]. There are two species, *S. vermiculatus* and *S. baileyi* (the latter is sometimes regarded as a variation of the former by some authors) in the family. The plants of *Sarcobatus* are branching stiff spiny shrubs endemic to the alkali plains and deserts in the Great Basin and the southwest deserts in western North America [33,34,35,36]. They are widely used for household equipment, fuel, construction, weapons, food, forage, and music instrument. Furthermore, probably due to its unusual habitat, the leaves of *S. baileyi* are pulpy, and, more importantly, its male inflorescence is cone-like [37]. Although the female flowers of *Sarcobataceae* were studied in the 19th century [38], little is known of their male inflorescences other than that their male flowers are arranged beneath peltate scaly bracts, have a very short filament, and have long anthers [30,37]. This poorly understood male inflorescence/flower is the focus of the current study.

If we restricted our study to male flowers in angiosperms and without any control, our study would have limited significance with regard to plant evolution. Male cones in conifers protect their pollen sacs under their lateral appendages, and ovulate cones in Pinaceae may protect their seeds after pollination [39]. What happened to the female cones in Cupressaceae remains poorly documented. Therefore, we chose an ovulate cone of *Cupressus sempervirens* (Cupressaceae) for the purpose of comparison.

## 2. Materials and Methods

The *S. baileyi* material studied was collected from a bush on the south shore of Mono Lake, CA, USA, in June 2019 during a field trip organized by UCRS (https://napc2019.ucr.edu/field-trips-workshops#postmeeting accessed on 3 January 2023). The *Cupressus* material was collected from Mallorca, Spain in June 2016 during a field trip organized by the University of Vigo. Both taxa are listed as Least Concern (LC), which means that no conservation actions are needed as per IUCN. All field collections of plant materials complied with relevant institutional, national, and international guidelines and legislation. Field observations indicated that male inflorescences of *S. baileyi* and ovulate cones of *C. sempervirens* were uniform in dimensions; therefore, a minimal collections (two organs) was made. Observations and photographs were made with a Nikon SMZ1500 stereoscopic microscope at the Nanjing Institute of Geology and Palaeontology, Nanjing, China. Micro-CT (micro computed tomography) of *S. baileyi* was performed using a Zeiss Xradia 520 at the Nanjing Institute of Geology and Palaeontology, Nanjing, China, while Micro-CT of *Cupressus* was performed using a Skyscan 1172 at the Institute of Botany, Beijing, China. The 3D reconstruction and virtual sections were generated using VG Studio MAX 3.0. The specimen of *S. baileyi* was fixed with FAA (formalin-aceto-alcohol), dehydrated, embedded in paraffin, sliced into 16 μm thick sections using a Leica RM2135 rotary microtome, followed by dewaxing twice in xylene for 20 min each time and then dehydrating through a graded ethanol series (in 100% twice, once each in 90%, 85%, and 70%) for 30 min each time. Then, the sections were stained with Safranin O and Fast Green. Sections were examined and digitally photographed on a Leica microscope (DMLB) equipped with a video camera (Leica, DC 300F; Leica Microscopy and Systems GmbH, Wetzlar, Germany). The materials were dried in open air before observation. SEM observation on pollen grains of *S. baileyi* was performed using a TESCAN MAIA3 SEM (scanning electron microscope) housed at the Nanjing Institute of Geology and Palaeontology, CAS. Our measurements were performed based on images of pictures made during Micro-CT, SEM, stereomicrography, and light microscopy. The paraffin sections were deposited in the School of Life Sciences, Northwest University, Xi’an, China. All images were recorded in TIFF or JPEG format and organized together using Photoshop 7.0 for publication.

## 3. Results

*Sarcobatus baileyi* are bushes that are about 1~2 m tall and intricately branched, and their proximal branches are in contact with ground. Their leaves are clustered on cushion-like bases on older wood, and their blade is dull green to grayish green, usually terete, arcuate, 5–16 mm, and pubescent (*n* = 12) (Figure 1a). The male inflorescence is cylindrical in form, approximately 9 mm long (from the bottom to the top), and 3 mm in diameter (*n* = 2) with a central axis (Figure 1a,e). The central axis is approximately 0.4 mm in diameter, tapering distally (*n* = 2) (Figure 1e). Peltate shields are helically arranged around the central axis, each with a stalk (approximately 0.1 mm wide basally and 1 mm long), as well as a terminal rhomboidal expanded shield (approximately 1.4 mm wide) with a central apophysis (*n* = 5) (Figure 1a,e). Male flowers are reduced and equivalent to a single stamen, without any trace of perianth and gynoecium. A total of five to six caducous stamens are centered around the stalk of each peltate shield, attached to the central axis by 0.1 mm long filaments (*n* = 8) (Figure 1a,d–f). Each stamen is tetrasporangiate, approximately 0.8 mm long, 0.5 mm wide, and 0.4 mm thick (*n* = 7), and have 2 lateral pollen sacs that are fused when mature, dehiscing by valves (Figure 1b–f). Pollen grains are spherical, pantoporate, and 24 μm in diameter (*n* > 40) (Figure 1g).

The *C. sempervirens* plant we sampled is a tall tree growing in a farm on Mallorca Island in the Mediterranean Sea (Spain) (Figure 2a). The tree is tall, evergreen, and columnar in shape with blue-green needles (Figure 2a). The ovulate cones are spheroidal in form and 2.1 to 2.3 cm in diameter (*n* = 2) (Figure 2b–e). There are 7–8 peltate bracts attached to the central axis (Figure 2b–e). Each bract is about 10 mm long, 1 mm wide basally, and 16 mm wide apically with cavities (possibly containing resin) of variable sizes near the tip (Figure 2b–e). Numerous ovules are clustered in the space around the pedicels of the bracts and are sheltered by the bracts completely (Figure 2b–e). Ovules are slightly flattened, with two ridges, about 4 mm long, 3 mm wide, and 1~1.6 mm thick (*n* = 10).

## 4. Discussion

Sarcobataceae, Gieskiaceae, Phytolaccaceae, and Nyctaginaceae are clustered in a clade of Caryophyllales in APG IV [30]. One of the features unique to Sarcobataceae is that its flowers/inflorescences are unisexual, a feature unusual enough to distinguish Sarcobataceae from other families in the same clade [30]. This makes its male reproductive organs (inflorescences/flowers) incomparable to other Caryophyllales and even to other angiosperms. If the details of the fruit and pollen grains are ignored, *S. baileyi* appears more like a gymnosperm; its inflorescence is unisexual, its leaves are fleshy as seen in conifers, and, most interestingly, its male inflorescences are cone-like catkins (Figure 1a). This comparison is intriguing, and it may shed light on the homology and origin of the male inflorescence of *S. baileyi* (angiosperms). It is well known that pollen sacs are in naked clusters in Ginkgoales [40], Gnetales [41], and Caytoniales [42,43], while the pollen sacs are present and, thus, protected on the abaxial/adaxial surface of “microsporophylls” in Cycadales [43] and Coniferales [43,44]/Bennetttiales [43,45]. The cover and protection for the pollen sacs are provided by the scale-form “microsporophylls”, reducing the possibility of harmful attacks on the pollen sacs. It is curious that such a protection for pollen sacs appears lacking in angiosperms, which usually protect their pollen sacs using petals or tepals. The male florescence of Sarcobataceae documented here provides a rare exception and supplementation for the above generalization: pollen sacs/stamens/male flowers are sheltered by peltate shields in Sarcobataceae. With this supplementation, the pollen organs in gymnosperms and angiosperms seem to implement the same logic in their evolution: offspring development conditioning (ODC). ODC has recently been found to be an almost ubiquitous trend in the evolution of reproduction in sexually reproducing organisms (including all higher plants) [46]. Our new observation seems to bridge and narrow the gap between gymnosperms and angiosperms, which otherwise are thought to be widely separated.

Coverage and protection of pollen sacs are important insurance for the successful reproduction of plants; however, this insurance as an implementation of ODC is not restricted to pollen sacs alone. This rule applies to the female organs equally. Ovules enclosed before pollination are restricted to angiosperms [3,39]. Such an enclosure is a more apparent coverage and protection for the ovules, and besides protection, this enclosure supplies secured nutrition for the healthy development of ovules/seeds [3,39]. Therefore, angiospermy/angio-ovuly, just as the above protection for pollen sacs, is another way to implement ODC.

If ODC is such a ubiquitous trend in the reproduction of plants, one cannot help but wonder whether it is applicable to the female parts in gymnosperms. If we can find an example in gymnosperms, it will strengthen the ODC hypothesis. Relative to angiosperms, gymnosperms are famous for their naked ovules/seeds; however, this does not mean that gymnosperms do not care for their ovules/seeds. Instead, gymnosperms implement protection equally, although in diversified ways. At least some conifers tend to seclude their seeds from the exterior after pollination through cone closure, which is implemented by enlargement and intercalary growth of bract bases [39], while the ovules/seeds of Bennettitales are protected by peltate interseminal scales with only their micropylar tubes exposed [47,48,49] and sometimes by contractile seeds [48]. It is noteworthy that the way Bennettitales protect their ovules is comparable to the method that Sarcobataceae apply to protect their male flowers/stamens/pollen sacs (Figure 1a,e), although they differ in the number of pollen sacs/ovules per peltate shield/interseminal scale. The ovules/seeds of *C. sempervirens* (Coniferales) (Figure 2b–e) are also sheltered by peltate shields in a way similar to those of Sarcobataceae (Figure 2b–e); the seed scales (former secondary fertile branches) are reduced to only clusters of ovules/seeds that are independent of the bracts, as seen in *Juniperus* [50,51]. Such a resemblance spanning remotely related major groups of seed plants is intriguing, and it may reflect a shared pattern underlying the evolution of reproductive organs across all seed plants. The occurrence of such a phenomenon in widely separated plant groups apparently cannot be due to phylogeny; it may well represent a convergence in evolution leading to protection of reproductive parts before pollination and anthesis, as well as parallel implementations of the same evolution trend, namely ODC.

Although various molecular studies [23,24,25,26,27,28] have been carried out to elucidate the gene network underlying flower development, male flowers in angiosperms (especially those of Sarcobataceae) appear understudied in molecular studies [52]. The organization of *Amborella* flowers has attracted much attention, and some conclusions have been published [53]; however, male flowers, such as those of *S. baileyi*, are left molecularly intact and poorly understood. Such a situation may be related to the huge gap between female parts in angiosperms and gymnosperms, which constitutes an obvious challenge in botany. We wish to call for more attention and studies to elucidate the molecular mechanisms underlying the development and evolution of the unique male reproductive organs of Sarcobataceae.

## 5. Conclusions

We documented the unique morphology and organization of male inflorescence of Sarcobataceae. Unlike in other typical angiosperm flowers, the male flowers of Sarcobataceae are sheltered under peltate shields until anthesis. Similar protection for reproductive parts occurs in various seed plants in different ways, either through sheltering using peltate-formed shields (as documented here), physical enclosing (angiospermy, angio-ovuly), or timed opening and closing of parts. These diverse methods seem to be connected by a common rule: offspring development conditioning (ODC).

## Figures and Tables

**Figure 1 plants-12-01917-f001:**
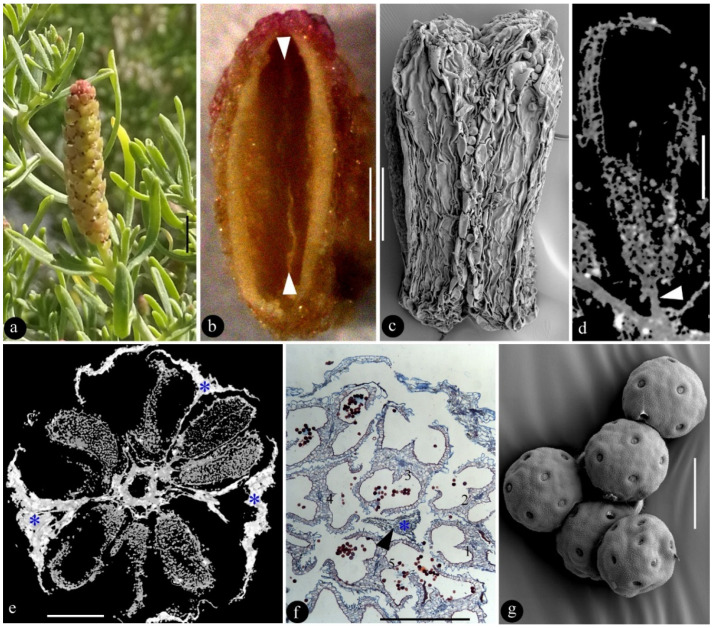
A male inflorescence of *S. baileyi* and its details. (**a**), a terminal upright male inflorescence. Scale bar = 2 mm. (**b**), lateral view of an opened anther. Note the residual ridge (triangles) between the former paired pollen sacs. Scale bar = 0.2 mm. (**c**), abaxial view of an anther. Scale bar = 0.2 mm. (**d**), Micro-CT virtual longitudinal section of a male flower attached to the axis by a short filament (triangle). Scale bar = 0.2 mm. (**e**), Micro-CT virtual cross section of the inflorescence, with male flowers (gray) and peltate shields (asterisks, white) arranged around the central axis. Scale bar = 0.5 mm. (**f**), tangential paraffin section of the male inflorescence showing five male flowers (1–5) centered around a shield stalk (triangle). Scale Bar = 0.5 mm. (**g**), pantoporate pollen grains. SEM. Scale bar = 20 μm.

**Figure 2 plants-12-01917-f002:**
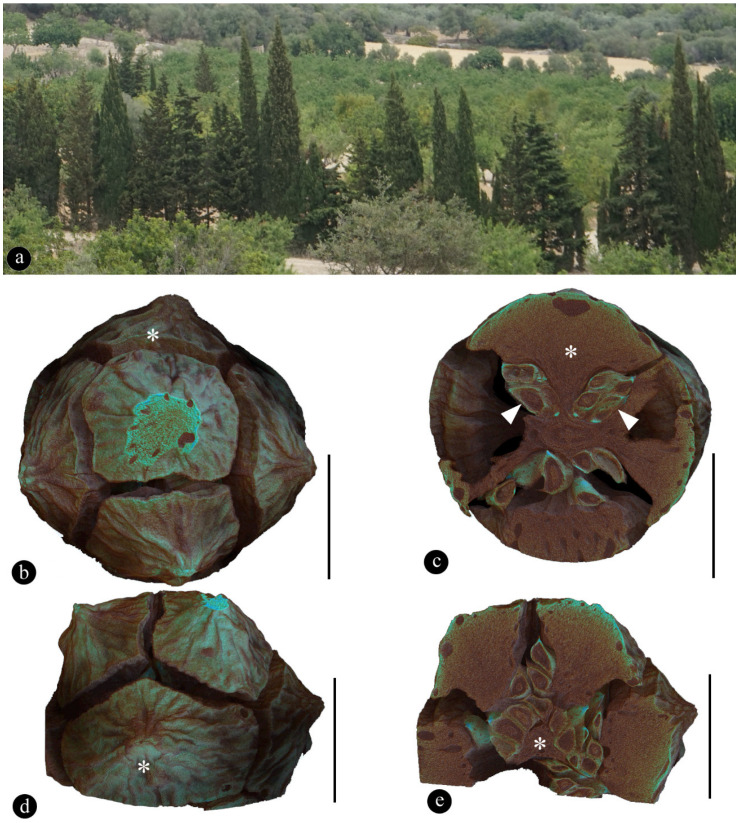
A female cone of *Cupressus sempervirens* (Cupressaceae). (**a**), a row of tall trees in a farm on Mallorca Island, Spain. (**b**), top view of the cone. Scale bar = 1 cm. (**c**), transverse section of the cone showing the top bract (asterisk) flanked by two groups of seeds (arrowheads), as orientated in (**b**). Scale bar = 1 cm. (**d**), side view of the cone. Scale bar = 1 cm. (**e**), tangential section cutting through the base (asterisk) of the bract, which is surrounded by many seeds, as orientated in (**c**). Scale bar = 1 cm.

## Data Availability

All data generated or analyzed during this study are included in this published article..

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
