# Peer review of "Unique Morphology of Sarcobatus baileyi Male Inflorescence and Its Botanical Implications"

_plants, 2023, doi:10.3390/plants12091917_

Round 1
Reviewer 1 Report (New Reviewer)
This is a kind of descriptive study, important from the point of view of plant systematics and evolution.It describes an interesting example of the plant of the transition between gymnosperms and angiosperms.
In general, the idea is nice. However, the study description is chaotic.
General comment – I can not understand, why do you use genus name (in the Introduction and Methods description) and the study results refer to Sarcobatus baile.
Moreover, explain why did you also studied Cupressus sempervirens (Fig 2).
Detailed comments
Key words- they should not repeat the words from the title, therefore replace the ‘male inflorescence', 'Sarcobatus'
Abstract
Abstract: A typical flower … You mean ‘A typical angiosperm flower…..’
L 16-18 s. Different from such a typical flower image, male inflorescence of Sarcobatus baileyi (Chenopodiaceae) has is more comparable to a cone in conifers: it lacks calyx, corolla, and gynoecium expected for a normal flower, and its male flowers are directly attached to a central axis and sheltered by a peltate indusium like shields.
Please, read this sentence carefully. It is completely strange. How a male flower it expected to possess gynoecium. ????
L 17 - has is more comparable : ‘has’ or ‘is’. There are such errors through the text. Please, correct the English language.
L 19 What does it mean ‘new nature’ – Explain ; Such a morphology reveals new nature of male flowers in angiosperms
In my opinion, the Abstract needs to be re-build. It did not give a key points of the study. Simply, try to indicate the flower features that are related to gymnosperms and angiosperms.
Also, indicate the novelty of the study for systematics and evolution.
Introduction
The aim of the study is poorly described. Try to explain, why do you undertaken the study.
In the results you described Cupressus sempervirens. Try to explain this in the study objectives.
L 52-53 This poorly understood male inflorescence/flower is the focus of current study’??? I need the explanation, why did you choose this species.
It is not clearly described the male and female flower traits in the genus Sarcobatus.
Methods
The authors described only microscopical procedures. It is good, however, I did not find the description of (i) how the flowers were measured, (ii) how much flowers was observed (n=????).
L 91_ If the details of the fruit and pollen grains are ignored, Sarcobatus appears more like
a gymnosperm: ….. It is quite strange statement. Please, re-phrase it. You can not ignore any traits of the plant species studied. Simply, try to describe the traits that are close to gymnosperm and angiosperm.
L 94 inflorescences approximately 9 mm long and 3 mm in diameter. How was measured (example - from base to the top??), how much inflorescences was measured; with digital caliper or so???
This notification is referred to all measurements described in this study.
L180 ….plants in different way’ Please, be more precise (what do you mean – indicate an example of ‘different way’).
This paper needs a lot of corrections before publishing
Reviewer 2 Report (New Reviewer)
The intention to study the flower morphology of the endemic species Sarcobatus baileyi is an interesting one, but the approach, methodology and results provided in this work can be significantly improved.
The abstract does not contain relevant information for the paper. Even the first sentence contains inaccuracies - there are many genera/species of angiosperms that do not have colored flowers and that lack the calyx or corolla (or both); a comparison with the flower/inflorescence of gymnosperms was not necessary.
In the Introduction, it is specified "The unisexuality of male flowers has been intensively studied [35]" - it cannot result from a single References paper that a structure is "intensively studied".
Although in the Introduction the authors talk about two species in the Sarcobataceae family - Sarcobatus vermiculatus and S. bailey (the latter being considered a subspecies by some authors), in the Materials and Methods Chapter appears "The Sarcobatus material studied...". What species/subspecies was investigated? In the explanations in Figure 1 we find the species Sarcobatus baileyi, but in the text, in the Results chapter, only the Sarcobatus genus is referred to.
The "Micro-CT of Cupressus" investigations are described in the Materials and Methods Chapter; but in the paper there are no such images for Cupressus. Also, for the SEM images in Figure 1, the method is not specified.
In the Chapter Results, different sizes for stamens, pollen grains etc. are mentioned, without specifying in Materials and Methods how these were obtained (if they are averages, from how many measurements, etc.).
The images used to describe the structure of the male inflorescence of Sarcobatus baileyi are too few and of questionable quality: for example, Figure 1 C shows an image (SEM) of an anther, which is significantly deformed due to the drying process (probably drying in open air instead of using CO2 critical point drying, for example).
In Figure 1 F, not many details can be observed. If paraffin sections were made (as mentioned in Materials and Methods), then images should be provided from different levels of the inflorescence, at different magnifications, in order to be able to observe structural details.
In the Results Chapter, only information about Sarcobatus genus is presented, not about Cupresssus. Images of female cone of Cupressus sempervirens are presented in Discussions!
The comparison between the male inflorescence of Sarcobatus baileyi and the female cone of Cupressus sempervirens is not convincing, and the information presented is not very relevant either.
This manuscript is a resubmission of an earlier submission. The following is a list of the peer review reports and author responses from that submission.
Round 1
Reviewer 1 Report
This is a short paper describing the staminate flowers of an endemic species of N America.
The Introduction consists of one paragraph and the role, importance and distribution of this species is not reported. There is not a section describing the rationale and aim of the research. The sampling has been made on one tree during one field trip, and the set of data seems poor.
Below, I remark some other arguments denoting important weak points in the discussion.
Abstract
The description with ‘4 whorls of organs’ cannot be presented as a ‘typical flower’ without specifying that this is the case of some families of entomophilous plants.
Line 16 = Sarcobatus; check this name and correct along the text
Introduction
Please, consider that Chenopodiaceae is no more a botanical family (APG IV, and previous editions), and Sarcobatus is in an own family. Actually the Authors mentioned the relevant paper, but did not report any information on why the genus was moved in a different family:
H.-Dietmar Behnke Sarcobataceae. A New Family of Caryophyllales, Taxon 46
WFO Plant List - https://wfoplantlist.org/plant-list/taxon/wfo-7000000546-2022-12
Methods
It is not clear why material taken from Cupressus is mentioned.
Line 35 = “The identifications were done by XW.”: this must be moved in the Author contribution section
Results
Line 61 = “Morphologically, the plant looks more like a gymnosperm” : this sentence is not justified by the morphological observations reported, and – for example – is not mentioned that this species produce fruits and the pollen is a multiporate pollen. This pollen morphology only exists in angiosperms.
Discussion
Line 81 = In angiosperms there are many examples of species/families with no attractive flowers. In this discussion the key roles of pollination strategies, the convergent evolution and phylogeny are not mentioned.
Line 98 = The ODC is mentioned without explaining what may be its role in the case study presented.
Putnam, H.M., Ritson-Williams, R., Cruz, J.A. et al. Environmentally-induced parental or developmental conditioning influences coral offspring ecological performance. Sci Rep 10, 13664 (2020). https://doi.org/10.1038/s41598-020-70605-x
Reviewer 2 Report
I understand some limitations of the 'short communication' form, but some information is missing both in the introduction and in the discussion section. Some brief description of the state of knowledge on the evolution of the flower within the angiosperms would have been helpful. The aim of the research presented is not fully outlined and I was not convinced why this research is ‘implications on flower forming’ (see title). Why do these research fit into the flower (evolution, develop ?) forming?
The same methods should be used to show the differences between male and female flowers, as well as between the studied species of Sacrobatus and Cupressus. It is not entirely clear why Cupressus sempervirens was chosen for comparison, due to its the closest phylogenetic relationship, or whether other characters were taken into account? Why did you not choose representatives of the ANA grade clade, which represents the oldest developmental lineages of angiosperms?
The evolution of flowers within angiosperms has been discussed many times (see references below), as well as molecular studies in floral formation, which shoul be mentioned in introduction section, and also describe in discussion.
Ma, Q., Zhang, W. and (Jenny) Xiang, Q.-Y. (2017), Evolution and developmental genetics of floral display—A review of progress. Jnl of Sytematics Evolution, 55: 487-515. https://doi.org/10.1111/jse.12259
Ludovico Dreni, Dabing Zhang, Flower development: the evolutionary history and functions of the AGL6 subfamily MADS-box genes, Journal of Experimental Botany, Volume 67, Issue 6, March 2016, Pages 1625–1638, https://doi.org/10.1093/jxb/erw046
Sauquet, H., von Balthazar, M., Magallón, S. et al. The ancestral flower of angiosperms and its early diversification. Nat Commun 8, 16047 (2017). https://doi.org/10.1038/ncomms16047
- Line 85-87 On what basis is this statement? no references what molecular study says. Inferring evolutionary relationships only on the basis of morphology without molecular data is incomplete, lack of a multidisciplinary approach may result in erroneous conclusions.
- Line 89 rarely or never?
- Line 95-97 lack of consistency in methodology, juxtaposistion of male and female flowers of both studied taxa
- What does the last sentence of the discussion (line 98-101) contribute to flower formation?
- References section, position 5 - font
Reviewer 3 Report
This manuscript presents a study of the male reproductive structure of Sarcobatus, and purpotes to make generalizations out of the findings. However, the manuscript is absolutely skeletal. Introduction, discussion, results, and figures are reduced to a minimum, with very few references. Moreover, the authors make odd interpretations of the structure, which is more clearly interpretable an inflorescence with bracts akin to a catkin, and even more oddly compare it to a female cone of a conifer. The manuscript should be substantially expanded to start to be even acceptable for publication, a more detail study of the structure needs to be included (with proper figures showing the anatomy and histology of the inflorescence), and a more expanded discussion that includes comparative discussion with other Caryophyllales should be added. The authors even assign Sarcobatus to Chenopodiaceae, while it has been show to be related to Nyctaginaceae and Phytolaccaceae. It seems like the authors have no interest in truly describing the structure, and instead focus on citing other works from one of the coauthors that is marginally relevant.
Reviewer 4 Report
The article describes the morphology of Sarcobatus male flowers. The authors explored the morphology of male flowers and discussed its implications on flower forming. Overall the article is interesting, but there are some points that need further clarification. Some considerations are described below in order to improve the quality of the manuscript:
Introduction:
The introduction can be improved. The authors can describe morphological aspects of floral species related to the Sacrobatus genus.
Material and methods:
In general, this section is good and well written.
Please spelling the abbreviations “Micro-CT” pg. 1, line 37, “FAA” pg.1, line 37.
Discussion,
Page 3, line 81. "Male flowers of Sacrobatus baileyi are atypical: neither beautiful nor attractive" and "has its stamens sheltered by peltate shields"
I guess two interesting points regarding to these statements that can be discuss are, first, what are the evolutionary traits that led to this floral morphology described in male flower of Sacrobatus baileyi and what are its ecological implications.